# Urban Heat Islands and Thermal Comfort: A Case Study of Zorrotzaurre Island in Bilbao

**Irantzu Alvarez** [1,*] , **Laura Quesada-Ganuza** [2] , **Estibaliz Briz** [2] **and Leire Garmendia** [2]

1   Graphic Design and Engineering Projects Department, School of Engineering, University of the Basque Country UPV/EHU, Building I, Plaza Ingeniero Torres Quevedo s/n, 48013 Bilbao, Spain

2   Mechanical Engineering Department, School of Engineering, University of the Basque Country UPV/EHU, Building I, Plaza Ingeniero Torres Quevedo s/n, 48013 Bilbao, Spain; laura.quesada@ehu.eus (L.Q.-G.); estibaliz.briz@ehu.eus (E.B.); leire.garmendia@ehu.eus (L.G.)

*   Correspondence: irantzu.alvarez@ehu.eus; Tel.: +34-946-014-996

**Abstract:** This study assesses the impact of a heat wave on the thermal comfort of an unconstructed area: the North Zone of the Island of Zorrotzaurre (Bilbao, Spain). In this study, the impact of urban planning as proposed in the master plan on thermal comfort is modeled using the ENVI-met program. Likewise, the question of whether the urbanistic proposals are designed to create more resilient urban environments is analyzed in the face of increasingly frequent extreme weather events, especially heat waves. The study is centered on the analysis of temperature variables (air temperature and average radiant temperature) as well as wind speed and relative humidity. This was completed with the parameters of thermal comfort, the physiological equivalent temperature (PET) and the Universal Temperature Climate Index (UTCI) for the hours of the maximum and minimum daily temperatures. The results demonstrated the viability of analyzing thermal comfort through simulations with the ENVI-met program in order to analyze the behavior of urban spaces in various climate scenarios.

**Keywords:** climate change; urban planning; resilience; heat waves; thermal comfort; simulation

## 1. Introduction

The fight against climate change has one of its principal battlegrounds in cities. This is the case not only because cities contribute in great measure to global warming, as the vast majority of resources are consumed within them and they release a large volume of emissions, but also because cities are very vulnerable to the effects of global warming and their own configurations can either accentuate or mitigate those effects.

One meteorological event that is accentuated due to climate change is the increasingly frequent appearance of heat waves of lengthier durations and higher intensities [1]. The impacts of these heat waves, although they vary from place to place, can be significant for the health of the population [2]. In addition, the way in which cities are constructed (e.g., the geometry and materials of the buildings and streets, the presence or absence of vegetation and water or the impermeability of the ground) can modify the local climatic conditions, prompting the appearance of local effects such as the heat island effect or the urban canyon effect, aggravating their impact in many cases.

Despite the magnitude of a climatic event, the seriousness of its impact will vary depending on the physical and socioeconomic conditions of the area, such as the geographic position, urban morphology and density and the governance system [3]. These factors play significant roles in the determination of the resulting impact, highlighting the anthropic contribution to events that are generally considered natural. For all those reasons, cities play a fundamental role in the establishment of strategies for the mitigation of the effects of climate change and in their adaptation [4]. In this framework, urban planning is a fundamental tool that permits local administrations to design urban spaces that are more resilient to climate change and future extreme weather events.

In the case of heat waves, thermal comfort within urban spaces is a fundamental factor [5]. Thermal comfort is the key indicator that describes the subjective temperature experience that each person has, combining the impacts of solar radiation and shade, wind, air temperature and relative humidity on thermal sensation. Thermal sensation is derived from thermal comfort and is key to a heat wave event. A human body at rest generates around 100 W of metabolic heat (as well as any absorbed solar heat), and if the ambient temperature is higher than the optimum central temperature of the human body (around 37 °C), the human body cannot dissipate heat [6]. On the other hand, sweating, the main process by which the human body regulates temperature, becomes less effective if the relative humidity is high, resulting in the accumulation of heat within the body and, therefore, an increase in morbidity and mortality.

## 2. Objective and Methodological Approach

The objective of this study is to analyze the impact of heat waves on thermal comfort in an open area as a function of its urban planning. To this end, a real case study was used: the North Zone of the island of Zorrotzaurre (Bilbao, Spain), where thermal behavior was studied in accordance with the urbanistic norms proposed in the special plan for its future construction [7] and its effect on external thermal comfort. This zone has yet to be constructed and, therefore, its modeling will help foresee problems and anticipate possible solutions before construction begins on-site.

The study was completed using the different modules of the ENVI-met program. A model of the zone was created with the SPACES module (the ENVI-met modeling module) following the criteria established in the Special Urban Plan for Zorrotzaurre, and a climatic simulation was run on ENVI-met to predict the thermal behavior of the exterior spaces during a heat wave period. The data introduced for the simulation were recorded at the weather station located on the island during a real heat wave (2016). In this simulation, the climatic parameters of the potential air temperature, relative humidity and wind were used, as well as other complementary parameters, such as the physiological equivalent temperature (PET) and the Universal Thermal Climate Index (UTCI), which helped understand thermal comfort in a clearer way.

## 3. Climate Change and Urbanism: Baseline Concepts for Thermal Comfort Assessment

According to the fifth evaluation report of the IPCC (Intergovernmental Panel on Climate Change), the average global temperature will have increased by 0.3–0.7 °C before 2035 and by 2 °C in 2100. If no change is produced, global warming and climate change might be inevitable, with temperature maximums found in urban centers and their areas of influence [1]. These rising temperatures, added to increasingly numerous heavy rainfall events and subsequent flooding, heat waves, rising sea levels and the increasing frequency of storms and other environmental disasters, represent serious challenges for cities. It highlights the need and the importance of urban planning that is aware of the inherent challenges of climate change in the development of adaptation strategies to move toward more resilient urban environments [8].

At present, there is no universal definition of a heat wave, but it is known that those extreme events associated with particularly high temperatures sustained over time produce notable impacts on human mortality, the economy and ecosystems [9]. Two well-documented examples are the Chicago heat wave of 1995 and the Paris heat wave of 2003 [10]. In both cases, high temperatures contributed to human mortality and provoked generalized and undesirable economic impacts and unease.

Even if the heat waves were characterized in a different way, such as according to the regional climate, the WMO guidelines on the definition and monitoring of extreme weather and climate events [11] define heat waves as periods of unusually hot and dry or hot and humid weather that can last for at least two or three days and can have a perceptible impact on human activities. The principal indicators for the characterization of heat waves are,

therefore, temperature and relative humidity; the level of relative humidity will define whether it is a dry or a humid heat wave [12].

The different regional administrations prepared studies to adapt the definition of a heat wave to the particular climatic characteristics of each region. Thus, the Basque government prepared a guide, in which the extreme temperature thresholds were established for the four climatic regions of the Basque Country [2]. This document established that Bilbao will form part of the Interior Cantabrian Zone. In the following table (Table 1), the extreme temperatures in this zone are shown to be over 36 °C, a threshold at which the lowest alert level alert is set. In addition, depending on the consecutive days for which extreme temperatures are reached, the minimum and the maximum daily temperature thresholds are set at 17 °C and at 35 °C, respectively.

**Table 1.** Extreme high temperatures defined for the climatic zone of Bilbao. The colors indicate the different levels of alert.

| | | Prevention Alert (Temperature Threshold in °C) | | |
|---|---|---|---|---|
| Interior Cantabrian Zone | Bilbao | Yellow ≥36 | Orange ≥38 | Red ≥40 |
| | | Temperature Thresholds (°C) | Alert as a Function of Consecutive Days | |
| | | Tmin ≥17 — Tmax ≥35 | Yellow 1–2 — Orange 3–4 — Red >5 | |

Existing studies on heat waves and their effects have demonstrated that as the morphological layout of an urban area affects both shade and ventilation, combined with the characteristics of the urban elements and the type and distribution of green spaces, heat waves will affect cities in specific ways, creating the heat island effect [13]. The geometry and the urban materials influence wind flows, energy absorption and the surface reflectance properties [14], which in turn conditions the effect and seriousness of the heat island. The effects of these drivers can be confirmed by observing the changes of the different parameters (e.g., temperature, humidity and PET) in the simulation based on constructive parameters, which include the differences in, for example, the built environment density, proximity to the river estuary and green zones of the study area.

## 4. The Case Study of Zorrotzaurre

### 4.1. Historical Development of Zorrotzaurre

Zorrotzaurre is the name of an artificially engineered island located at the extreme northeast end of the city of Bilbao (Spain) (Figure 1), situated between the peaks of Elorriaga (to the northeast) and Kobeta (to the southeast), where the city gives way to the metropolitan area and the river estuary opens to the sea. The island has a surface area of 500,000 m$^2$, a length of 2 km and is approximately 250 m wide, creating a lengthy, narrow island in the NW/SW direction in the middle of the estuary. It is located on the alluvial plain of the Nervión-Ibaizabal River, close to the river's outlet into the Cantabrian Sea (approximately 7 km from the sea), which was converted into an island after a long process of transformation. This reconversion of the island was the great urban change foreseen in the city after a period of urban regeneration that took place during the 1990s. In Zorrotzaurre, a new area of opportunity is unfolding that lays the groundwork for the urban model of Bilbao and its expansion, and within it, it may be seen how the city of Bilbao is responding to the challenges of the current climate crisis.

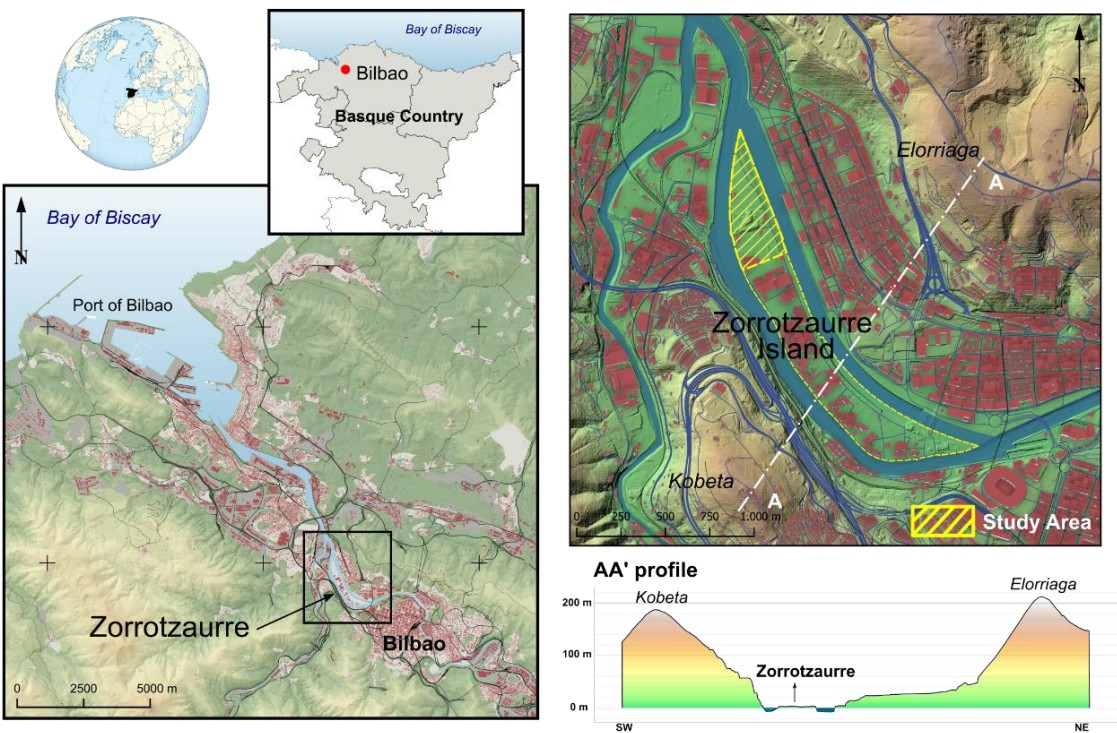

**Figure 1.** Location of the island of Zorrotzaurre and the study area.

Zorrotzaurre has undergone a large-scale transformation throughout the 20th century (Figure 2) that has substantially modified the layouts of both the city and the metropolitan area [15]. It is an area that is closely linked to the estuary, emblematic of the city and all of its urbanistic history.

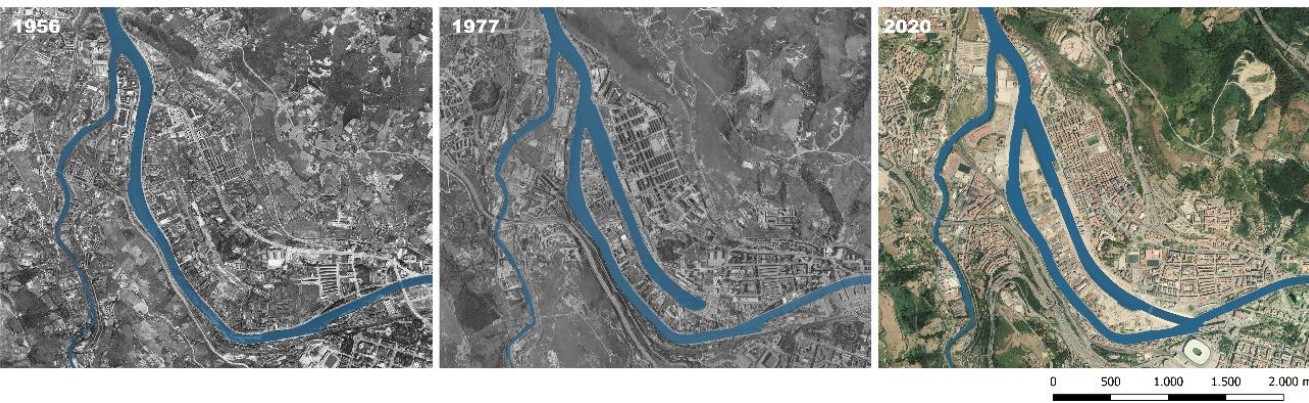

**Figure 2.** Evolution of the area as a meander of the river, the construction of the Deusto Canal and the creation of the island of Zorrotzaurre. Source: Spatial Data Infrastructure of the Basque Government (www.geo.euskadi.eus).

At the beginning of the 20th century, the area was a fertile alluvial plain between the meanders of the river at the outskirts of the city on the right bank of the river estuary, where the boats might run aground on sandbanks on their way toward the center of the city on the same curve where the island is now located. There was already a busy industry located on the banks of the river estuary [16,17]. A plan had previously been proposed in 1929 (Urban Extension Plan of 1929) in response to those problems to open a canal (see Figure 2) and thereby enlarge the maritime and commercial potential of the city. The installation of significant industry proliferated in the area over those years. Work on the canal commenced in 1950, and it began to function as a dock in 1968. However, a little time afterward, the dock activity was moved to the seaport on the coast, and the mooring

was used far less (although it was still operative as a dock until 2006). The crisis of the 1980s meant that the idea of the island had turned into a peninsula before a long process of industrial degradation and urban decline commenced. At the end of the 1980s, the idea was floated of a plan to regenerate the zone, but large-scale urban regeneration works were undertaken within the city (the construction of the Guggenheim Museum and the metro, among others), which meant that the Zorrotzaurre project was left in the background.

In 2002, the management committee for the rehabilitation of Zorrotzaurre was set up, and in 2004, the first master plan, the work of the architect Zaha Hadid, was presented [7]. In this plan, a large island was proposed with ten bridges, high-rise buildings for housing, offices and cultural services. After some years of reflection and change, and after the crisis of 2007, a new special plan for Zorrotzaurre was presented in 2012 with more green zones, more pedestrian areas and bicycles and in which there was more room for regeneration and reutilization of spaces. This first plan incorporated the complete opening of the canal, turning the peninsula into an island.

In 2014, work recommenced to finish the canal, which had begun in the middle of the preceding century and which was finished in 2018. In 2020, work on the urbanization of the first phase of the construction of the new island started, which was to become the North Zone, where the first 600 homes were to be built. A new phase started here which following the pattern of earlier phases and confronted another large-scale crisis, this time in the form of climate change.

In addition, specific and unique climatic circumstances arose in this project that made it appropriate for the study of microclimatic modifications generated by the construction of urban infrastructure. Its location within a valley with a NW/SE orientation exposed to gusts of wind that were frequent in the zone and the lack of green zones with trees, together with the lack of any earlier urban infrastructure, constituted a site that was under construction and suitable for modeling and study.

Following the long process described above, the island of Zorrotzaurre might be seen as the touchstone of urbanism in Bilbao. The island has, after a period of oblivion, assumed a fundamental role in the transformation of Bilbao, providing a response to social, economic and environmental needs through sustainable and resilient construction, creating future urban environments adapted to the needs and challenges associated with climate change.

### 4.2. The Climate in Bilbao

The climate in Bilbao and its metropolitan area is, due to its proximity to the sea, a humid ocean climate. A temperate climate prevails throughout the year with mild winters and hot summers, and only on some days of the year are temperatures more severe (i.e., over 30–35 °C or below 0 °C. The average temperatures fluctuate between 10 °C during the winter months and 20 °C in the summer months. Rainfall is present throughout the year, with average annual values over 1100 mm, although with no dry season, the driest months occur in summer, and the wettest months occur in winter. The prevailing winds are from the southeast (Figure 3).

The weather station of Deusto, situated at the extreme northern end of Zorrotzaurre, was selected for the analysis of the climatic data of the area under study (Figure 4).

According to the data from the Deusto weather station, over the past decade, there have been four episodes of heat waves with extreme consecutive temperatures (for two days in all cases). All these episodes occurred during the months of June and July, except for the events of 2018, which took place in October. From these episodes, the heat wave recorded on 18 and 19 July 2016 was selected because it reached one of the highest temperatures ever recorded for that season (40.4 ℃, considered a red alert in the Basque early warning system), and the minimum temperature was never lower than the threshold value of 17 °C marked for the zone (the minimum of the two days was 20.8 °C). This heat wave was due to a mass of very warm air from the African continent that covered much of the Iberian Peninsula.

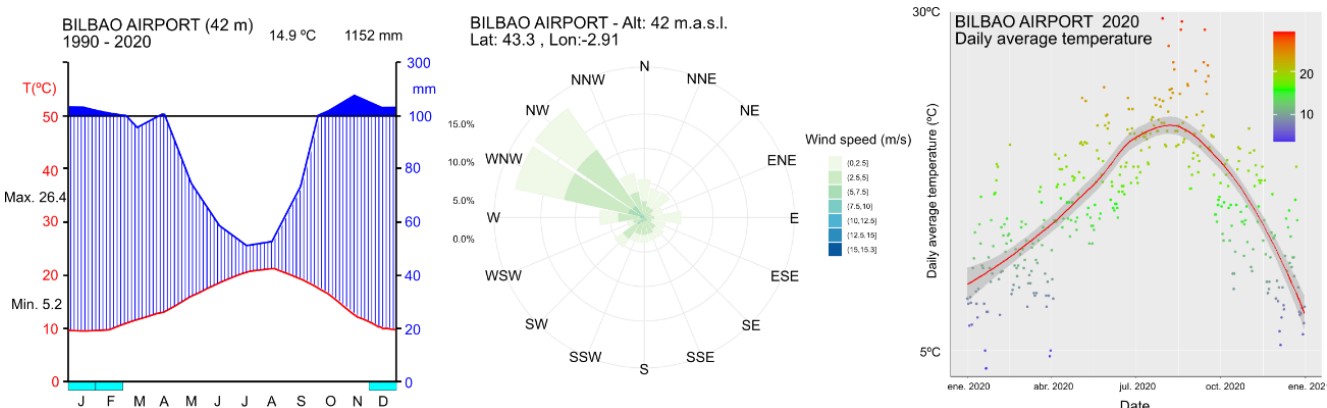

**Figure 3.** Meteorological data from Bilbao (climograph, wind rose and graph showing average daily temperature data for 2020). Source: Spanish Meteorological Agency (AEMET).

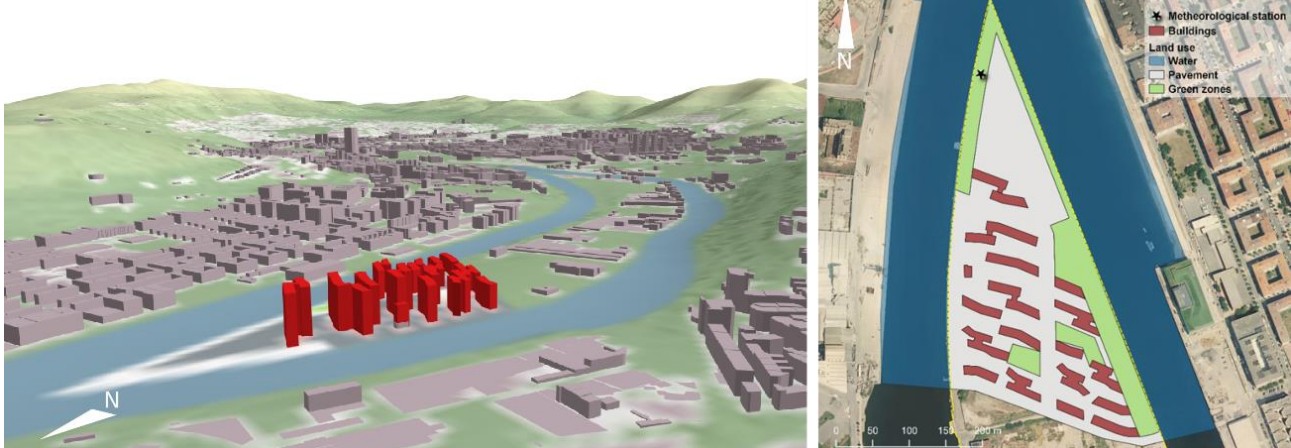

**Figure 4.** 3D model of the North Zone of Zorrotzaurre (**left**) and the land usage and buildings (**right**) introduced in the modeling process.

## 5. Evaluation of Thermal Comfort in Zorrotzaurre

### 5.1. Initial Data for Modeling

The data on the buildings to be constructed in the North Zone of the island, according to the Special Urban Plan of Zorrotzaurre [7], were extracted for modeling purposes. The model included, on the one hand, the permitted usage of the land in each zone and subzone (e.g., residential, tertiary and equipment) and the basic urbanistic norms with which urban development and future buildings must comply. Among them were the minimum separation between buildings, the length of opposing facades, the height of each building and the number of stories.

All this information was included in a detailed plan that was digitalized and from which the shapes and heights of future buildings were represented (Figure 4), as well as the varied land usage within the zone, differentiating between asphalted and green zones.

### 5.2. Modeling

The ENVI-met program version v.4.4.5 was selected for the modeling and simulation, preparing a micro-scale model with which to analyze the biometeorological conditions and their changes during a heat wave period. ENVI-met software can be used to simulate climates in urban environments and to evaluate air conditions, vegetation, architecture and construction materials and their effects on the environment and thermal comfort [17] (Acero and Herranz-Pascual, 2015). It is a tri-dimensional urban microclimate model designed to

simulate surface–plant–air interactions, with a standard spatial resolution of 0.5–10 m and a photogram time of 10 s. It permits the high-resolution analysis of heat and energy flows.

In the ENVI-met modeling process, the simulation parameters (microclimate data and construction elements) were modeled and edited using the database of materials to create the area input files (.INX), the simulation file (.SIM) and finally the results evaluation archive (.EDT/.EDX) with BIOMET after the simulation and its visualization (LEONARDO).

The study was centered on the air temperature and average radiant temperature variables, as well as the wind speed and relative humidity for the analysis of thermal comfort within the chosen zone:

- Air temperature (°C): evaluates the distribution of temperature in relation to the influence of building layout and shading effects, as well as the properties of the materials used to pave the area and the eventual influence of the wind on specific parts of the zone that is under study;
- Wind or air speed (m/s): evaluates the distribution of the wind or air;
- Wind direction (°): evaluates the orientation of the wind or air;
- Relative humidity (RH) (%): evaluates the effect of the relative humidity and its temperature-related impact, combined with the principal directions of ventilation.

These data were obtained from the records of the Deusto weather station on 18 July 2016 and were input into ENVI-met in a CVS file that contained data recorded every 30 min between 12:00 a.m. and 11:30 p.m.

The area of the model was composed of a 96 × 122 × 30 cell mesh (in an x, y, z tridimensional reference system) with a cell size of 4.75 × 4.75 × 4.5 m (Figure 5). The materials chosen for the model can be seen in Figure 5, consisting of concrete pavement (in medium gray), asphalt in the zones for rolling traffic (in black) and the default brown color of the program for the permeable sandy terrain, with dense vegetation shown in green.

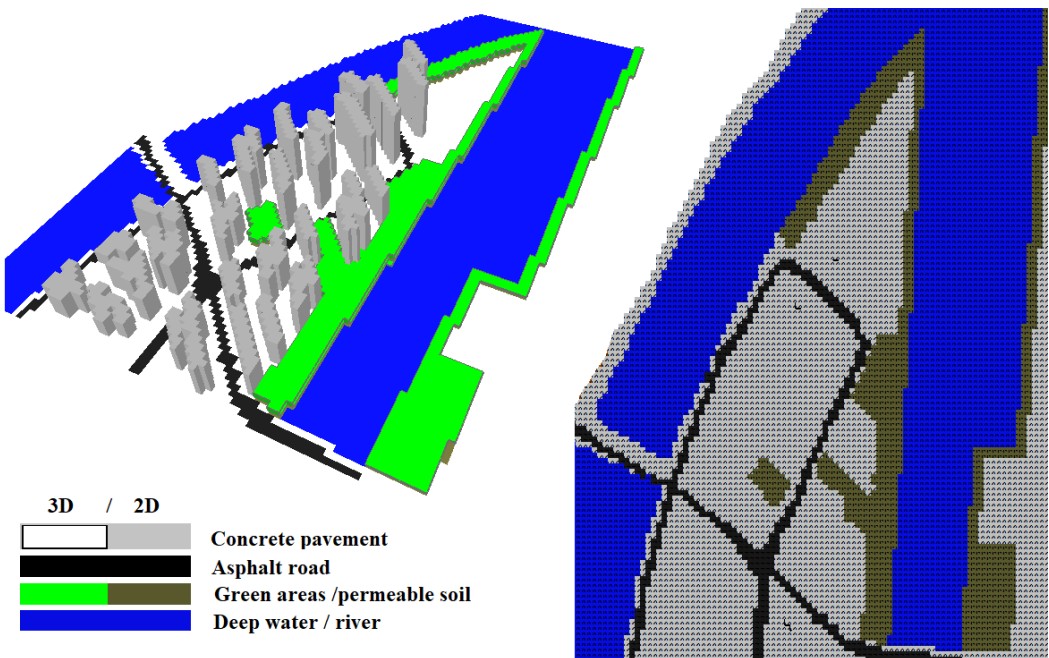

3D / 2D

☐ ☐ **Concrete pavement**
■ ■ **Asphalt road**
■ ■ **Green areas /permeable soil**
■ ■ **Deep water / river**

**Figure 5.** 3D and 2D models in ENVI-met SPACES, showing the Building and green zone components of the model (**left**) and the materials and surfaces (**right**).

The ENVI-met program version v.4.4.5 was selected for the modeling and simulation, preparing a micro-scale model with which to analyze the biometeorological conditions and their changes during a heat wave period. ENVI-met software can be used to simulate climates in urban environments and to evaluate air conditions, vegetation, architecture and construction materials and their effects on the environment and thermal comfort [18].

It is a tri-dimensional urban microclimate model designed to simulate surface–plant–air interactions, with a standard spatial resolution of 0.5–10 m and a photogram time of 10 s. This permits high-resolution analysis of heat and energy flows.

In the ENVI-met modeling process, the simulation parameters (microclimate data and construction elements) were modeled and edited using the database of materials to create the area input files (.INX), the simulation file (.SIM) and finally the results evaluation archive (.EDT/.EDX) with BIOMET after the simulation and its visualization (LEONARDO).

The study was centered on the air temperature and average radiant temperature variables, as well as the wind speed and relative humidity for the analysis of thermal comfort within the chosen zone:

- Air temperature (°C): evaluates the distribution of temperatures in relation to the influence of building layout and shading effects, as well as the properties of the materials used to pave the area and the eventual influence of the wind on specific parts of the zone that is under study;
- Wind or air speed (m/s): evaluates the distribution of the wind or air;
- Wind direction (°): evaluates the orientation of the wind or air;
- Relative humidity (RH) (%): evaluates the effect of the relative humidity and its temperature-related impact, combined with the principal directions of ventilation.

These data were obtained from the records of the Deusto weather station on 18 July 2016 and were input into ENVI-met in a CVS file that contained data recorded every 30 min between 12:00 a.m. and 11:30 p.m.

The area of the model was composed of a 96 × 122 × 30 cell mesh (in an x, y, z tridimensional reference system) with a cell size of 4.75 × 4.75 × 4.5 m. The materials chosen for the model can be seen in Figure 5, consisting of concrete pavement (in medium gray), asphalt in the zones for rolling traffic (in black) and the default brown color of the program for the permeable sandy terrain, with dense vegetation shown in green.

The thermal characteristics of these materials are as follows (Table 2).

**Table 2.** Thermal characteristics of the materials.used in the model (Figure 5).

| Material | Albedo | Emissivity | Conductivity | Color in Model (Figure 5) |
|---|---|---|---|---|
| *Pavement in gray concrete* | 0.5 frac | 0.9 frac | 1.63 W/(mK) | |
| *Asphalt* | 0.2 frac | 0.9 frac | 0.9 W/(mK) | |
| *Permeable soil* | 0.2 frac | 0.98 frac | Depends on water content | |
| *Water (river)* | 0 | 0.96 frac | - | |

After the simulation in ENVI-met, a subsequent analysis was conducted with BIOMET for the calculation of two complementary parameters that assist with better compression of the general thermal sensation for a person in the given scenario (thermal comfort): the physiological equivalent temperature (PET) and the Universal Thermal Climate Index (UTCI).

The PET is an index of thermal comfort that is based on a prognostic model of the human energetic balance, which includes calculations of the skin temperature, core temperature of the body, the sweat rate and, as an auxiliary variable, clothing temperature [19]. The model can be used for both a stationary and a non-stationary focus. In our case, BIOMET used the stationary solution of the corporal parameters because they are traditionally used in this sort of simulation when considering urban areas.

The Universal Thermal Climate Index (UTCI) is an indicator of thermal comfort based on body heat balance models which is designed to be applicable to all seasons and climates and for all spatial and temporal scales [20]. The UTCI is defined as the air temperature of the reference conditions that causes the same response in the model as in real conditions. The deviation of the UTCI of the air temperature depends on the real values of the air temperature and the average radiant temperature, wind speed and humidity, expressed as water vapor pressure and relative humidity [21].

## 6. Simulation Results

The times of day when the maximum and the minimum temperatures were recorded were chosen for the analysis: at 4:00 p.m. with a maximum of 40.4 °C and at 4:00 a.m. with a minimum of 20.84 °C. The air temperature, relative humidity, wind speed and average radiant temperature were analyzed at those times of day (Figure 6).

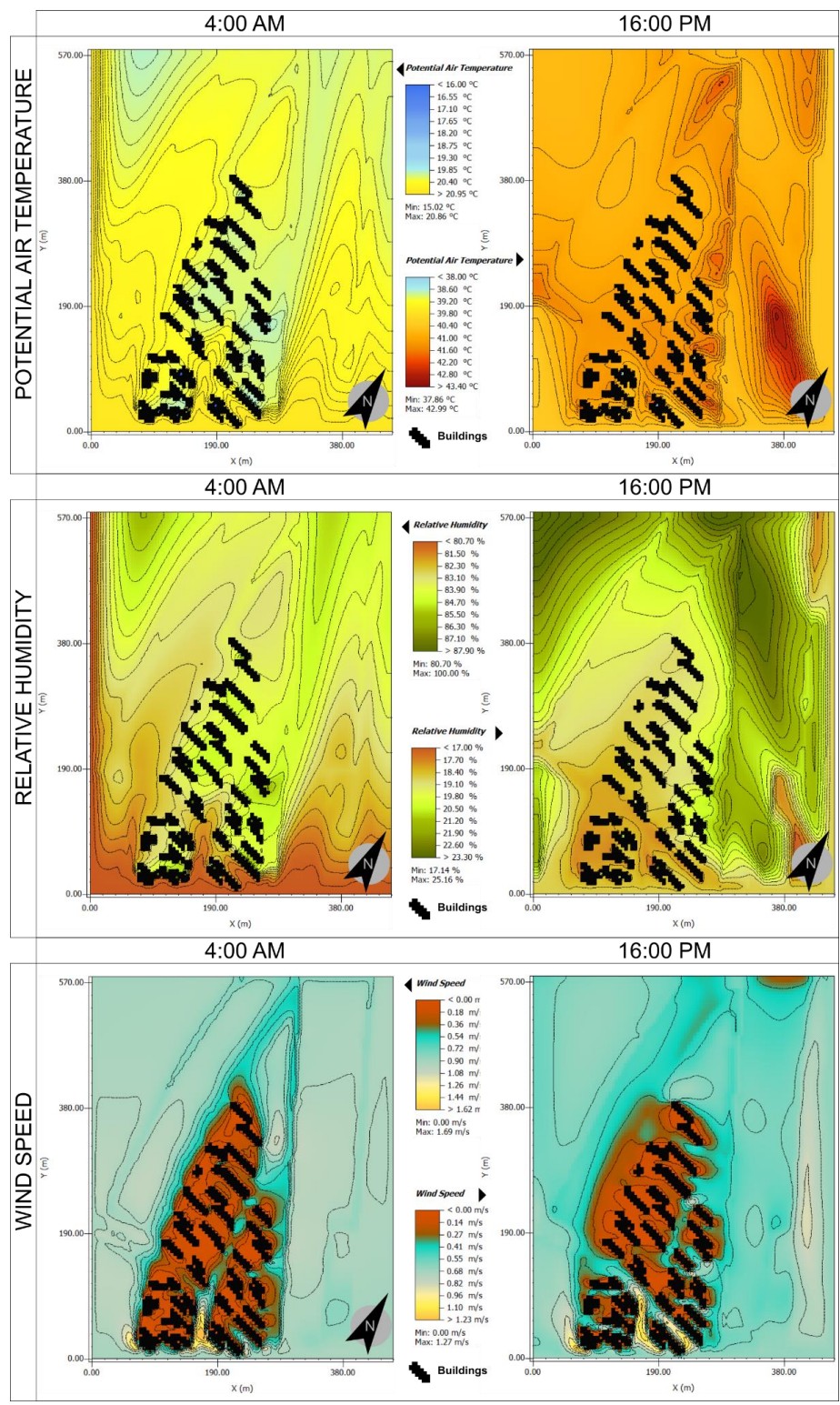

**Figure 6.** Modeling results (potential air temperature, relative humidity and wind speed).

The analysis of the results was focused on the data generated in the area of the island, because any measurements generated over the water or close to the limits of the model might not have been exact [22]. Likewise, the results that showed the formation of heat islands were highlighted [23].

In the case of the potential temperature of the air, the results can be seen in the first two images of Figure 6. At the time of the minimum temperature (4 a.m.), the minimum values were situated to the east of the island in the zones close to the green area covering that part of the island, as well as in the green zone between the buildings to the northeast. The maximums in this case arose in the central zone of the model, where the road crosses the island connecting the bridges. The difference between the maximum and minimum was significant, being higher than 5 °C. In the case of the potential air temperature at 4 p.m., an area of higher temperatures can be observed on the main road and around the buildings closest to the green zones.

In the case of relative humidity, the maximums were at around 4:00 a.m. and 4:00 p.m. in the green zones, both to the east and the north of the island, as well as in the green zone situated between the buildings. The minimums were situated on the main road that connects the bridges, a minimum that was very notable at 4:00 p.m.

The behavior of the wind in the simulation was quite similar at both hours of the day. As was previously mentioned, the prevailing wind in this zone is northeasterly, and for that reason, the maximums were given on both sides of the island parallel to the river estuary. However, the wind speed minimums occurred in the spaces between the buildings, principally in those with greater densities of buildings.

The PET and the UTCI were extracted from the subsequent BIOMET analysis (Figure 7). These values were calculated for the same hourly sections, and their comparison found that the PET achieved a finer resolution for the scenario due to the distribution of the urban network. The areas with no UTCI results appear in white.

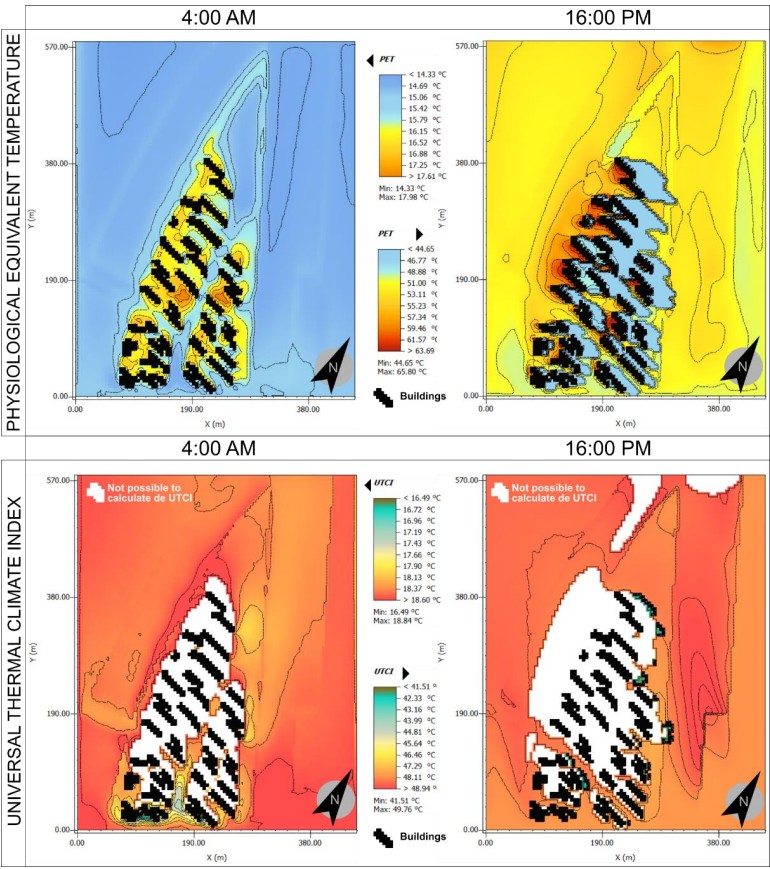

**Figure 7.** Modeling results (PET and UTCI).

With respect to the PET, it was the parameter that contributed the most information. The temperature maximums were noted in areas with a higher density of the built environment and less presence of green areas, and the incidence of solar radiation was of great importance. The minimums were noted during the night in zones with higher densities of green areas as well as lower incidences of solar radiation.

## 7. Discussion of the Results

From an analysis of the results, it may be clearly seen that the positions of the buildings, perpendicular to the river estuary toward the northeast, served as wind screens. The wind entered mainly from the northeast, which is relevant because, as mentioned in Section 4.2, it is a zone with highly frequent gusts of wind.

As a negative counter effect of this arrangement of the buildings, there is less ventilation in the narrowest streets that are parallel to the buildings. This means that during heat waves, the heat accumulates in zones of high built environment densities and less wind, creating an urban canyon effect to some degree (see the wind speed in Figure 6). This effect is especially appreciated at nighttime, as can be seen from the PET at 04:00 a.m., since a degree of heat accumulates in the least ventilated spaces between buildings.

The importance of the green zones in the event of a heat wave can be observed in the difference in relative humidity which, despite having been simulated with simple zones of grass and without any more vegetation, was much higher in those zones. This difference could be comparatively observed because the two first lines overlooking the river, both east and west, behaved in different ways depending on the presence of the green zones. It is also very notable that the zones with higher air temperatures coincided with the areas with no green zones, such as the road that runs along the length of the island and connects the bridges, where the relative humidity was lower. This area was simulated in accordance with what was mentioned in the previous section, with medium gray concrete pavement and asphalt in the zone around the main road without areas of vegetation or permeable soil.

It is significant to note that the streets that were shaded during the day, even though they were zones of high air temperatures, had lower PETs due to the lower solar incidence, but at night, without the effect of the sun, they were turned into heat island zones, where the PETs were higher.

The maximum PET at 4:00 p.m. was noted on the east coast of the island, a zone proposed with neither green areas nor shading and with a high solar incidence.

In the case of the UTCI, the simulation was not conclusive, given that the high density of the urban area prevented the calculation. The only relevant conclusion was its lower value in the zones with green spaces to the east of the island compared with the west zone, which was modeled as concrete pavement and non-permeable ground.

## 8. Conclusions

This study has demonstrated the climatic behavior of the proposed urban plan for the northern point of Zorrotzaurre during a heat wave and how urban design can affect the global thermal behavior of the zone.

Regarding the orientation of the buildings, the perpendicular orientation to the prevailing winds means that the incidence of the wind, very relevant in this zone, can be reduced. Nevertheless, in the case of high temperatures, the ventilation of the zone and the capability to dissipate heat during the night in the areas of the highest built environment densities is penalized, thereby creating urban canyon effects.

The effect of the green zones and the high value of the temperature in the area around the main road demonstrates that materials with high thermal inertia and the absence of zones of permeable soil worsen thermal comfort, driving the urban canyon effect during the night.

An increase in the shaded zones, especially on the east coast of the island, as well as more green and permeable soil areas could potentially improve thermal behavior in the areas with the maximum PET values.

Finally, it may be highlighted that the climatic parameters and comfort indices employed in this study are valid for the study of the incidence of heat waves in urban areas and could also be used for the study of alternative constructive solutions, thereby permitting an objective comparison of the results. This study has therefore demonstrated the relevance of urban planning in relation to the impact of extreme events such as, in this case, heat waves.

**Author Contributions:** All authors contributed equally to this work. All authors have read and agreed to the published version of the manuscript.

**Funding:** This research was funded by research group IT1314-19 of the Basque Government.

**Institutional Review Board Statement:** Not applicable.

**Informed Consent Statement:** Not applicable.

**Data Availability Statement:** The meteorological information used has been downloaded from the Open Data page of the Basque Government (https://opendata.euskadi.eus, accessed on 26 May 2021) and the spatial information from the Spatial Data Infrastructure page of the Basque Government (https://www.geo.euskadi.eus, accessed on 26 May 2021).

**Conflicts of Interest:** The authors declare no conflict of interest.

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
