# Peer review of "Urban Heat Islands and Thermal Comfort: A Case Study of Zorrotzaurre Island in Bilbao"

_sustainability, doi:10.3390/su13116106_

Round 1

Reviewer 1 Report

Well-organized argument. The findings are presented clearly. I appreciate the figures.

Please explain some of the abbreviations (e.g., SPACES on 67, IPCC on 77).

Heat Inland effect, Urban Canyon effect (on 36-37) -- no need to capitalize? 

Author Response

We have explained the meaning of the abbreviations SPACES and IPCC.

We have put the names urban canyon and heat island in lower case letters.

Reviewer 2 Report

This paper is confirmationary rather that groundbreaking. I am not convinced it does more than confirms by another means what is more or less known already.

A couple of potential improvement could be in the presentation of the modelling outputs which could be made clearer in two ways. It's not entirely clear from the images themselves what the black (showing buildings) and it is not entirely clear what the white areas display in Figure 7 for the UTCI graph. But this is all relatively clear in narrative.

Author Response

All the comments have been addressed.

We have added new items in the legend of the maps showing the modelling results to explain the buildings and white areas.